

# Nanoparticles modified bioceramic sealers on solubility, antimicrobial efficacy, pushout bond strength and marginal adaptation at apical-third of canal dentin

Basil Almutairi and Fahad Alkhudhairy

Department of Restorative Dental Sciences, College of Dentistry, King Saud University, Riyadh, Saudi Arabia

Corresponding author
Fahad Alkhudhairy,
Falkhudhairy@ksu.edu.sa

## ABSTRACT

**Objective**. The present study investigates the solubility, antimicrobial potency, pushout bond strength (PBS), and marginal adaptation of sealer modified using various nanoparticles (NPs) (silver, chitosan, HapNPs) to the root dentin at the apical third. **Methods**. Forty-four human premolars were prepared for root canal treatment *via* a chemical mechanical approach. The teeth included were subsequently distributed into four groups based on the type of modified and unmodified sealers. Group 1 bioceramic (BC) sealer (Control), Group 2 (AgNPs modified BC sealer), Group 3 (CHNPs modified BC sealer), and Group 4 (HApNPs modified BC sealer). Solubility was assessed by making five samples and measured *via* precision balance. Antimicrobial testing against *E. faecalis* was performed using the Agar diffusion test. The bond strength and failure modes were assessed utilizing a universal testing machine and stereomicroscope respectively. A pair of specimens from each cohort was fixed to an aluminum stub to assess marginal adaptation *via* SEM at the apical third. Data analysis was performed utilizing one way ANOVA and followed by Tukey's *post hoc* test ($p < 0.05$).
**Results**. Sealers-modified HApNPs exhibited the lowest solubility ($3.884 \pm 0.15\%$) and strongest PBS ($9.63 \pm 0.12$ MPa). Group 1 (Control) demonstrated highest solubility rate ($6.144 \pm 0.39\%$) and weakest bond strength ($6.50 \pm 0.09$ MPa). All the modified sealers exhibited the largest zone of inhibition with no significant difference. Whereas the unmodified BC sealer group presented the smallest zone of bacterial inhibition.
**Conclusion**. Marginal adaptation of sealers does not affect the bond strength outcomes achieved. The incorporation of silver, chitosan, and hydroxyapatite nanoparticles into BC root canal sealers resulted in reduced solubility, enhanced antibacterial efficacy, and improved PBS compared to unmodified BC canal sealers.

## INTRODUCTION

Root canal treatment (RCT) is a frequently performed dental procedure, involving the extirpation of diseased pulp tissue followed by sealing of the canal to prevent reinfection

(*Estrela et al., 2014*). Bioceramic (BC) root canal sealers have gained widespread popularity due to their biocompatibility, superior sealing properties, and high alkaline pH (*Lee et al., 2017*). However, these sealers are not particularly effective against *Enterococcus faecalis* (*E. faecalis*), a bacterium notorious for causing persistent endodontic infections and contributing to treatment failures (*Siqueira & Rôças, 2008*). Additionally, the bond integrity between root canal sealers and dentin is a critical factor in ensuring that the root-filling material remains securely attached to the canal walls during restorative procedures or normal chewing, thus preserving the integrity of the seal and the overall success of the treatment (*Santos-Junior et al., 2019*). As a result, researchers and dental professionals are continually investigating ways to enhance the antimicrobial properties of endodontic sealers while improving their mechanical performance.

The field of nanotechnology has made remarkable strides in recent years, garnering attention from various scientific disciplines, including dentistry (*Bapat et al., 2019*). The application of nanomaterials in endodontics is becoming increasingly evident, as reflected by the growing body of research on the subject (*Jandt & Watts, 2020*). Notably, silver nanoparticles (AgNPs) are widely recognized for their potent antibacterial properties and are commonly incorporated into dental composites, endodontic sealers, and medicaments (*Bruna et al., 2021*). Past research has consistently established that even at low concentrations, AgNPs significantly enhance the antibacterial efficacy of dental materials (*Mouafy et al., 2023*; *Yin et al., 2020*). Furthermore, these NPs have demonstrated no detrimental impacts on the mechanical qualities of root canal sealers or medicaments, nor do they induce tooth discoloration (*Afkhami et al., 2024*; *Suzuki et al., 2018*).

Another nanofiller that has garnered noteworthy attention in dentistry due to its various beneficial properties is chitosan nanoparticles (CHNPs). Chitosan is a safe, organic polysaccharide obtained from the deacetylation of chitin present in the exoskeletons of crustaceans (*Rampino et al., 2013*; *Shi et al., 2006*). It has demonstrated excellent antimicrobial potency, particularly against *E. faecalis* and Candida *albicans* (*Del Carpio-Perochena et al., 2015*; *Wieckiewicz et al., 2016*). In a lab-based study, *El-Tayeb & Nabeel (2023)* found that CHNPs and silver nanoparticles (AgNPs) did not exhibit superior antimicrobial efficiency when incorporated into bioceramic (BC) sealers. Additionally, hydroxyapatite nanoparticles (HApNPs) have found widespread applications in both medicine and dentistry, gaining recognition for their remineralizing potential, strong antibacterial properties, and solubility (*Wang et al., 2021*). The bactericidal effect of HApNPs is primarily due to their rapid ion release and generation of reactive oxygen species (ROS) (*Charpentier et al., 2012*). In the past, these NPs have been integrated into epoxy resin sealers to modify their properties, with results showing no significant differences from the control (*Guo et al., 2022*). However, there are no existing studies that have explored the potential benefits of HApNPs in terms of antibacterial efficiency, bond strength, and solubility when added to BC root canal sealers, highlighting the need for further investigation.

Based on the available literature, the present contemporary study investigates the antimicrobial potency, solubility, PBS, and marginal adaptation of sealer at the apical third of canal dentin when modified using various NPs (silver, chitosan, HapNPs).

Therefore, it was postulated that there would be no significant difference in the antibacterial effectiveness against *E. faecalis* when nanofillers modified BC sealers were used as compared with unmodified BC sealers. Additionally, it was also projected that the solubility, bond strength, and marginal adaptation of NPs incorporated BC sealer to the canal dentin would be non-inferior to that control. The null hypothesis would be that there is no significant difference in the antibacterial effectiveness against *E. faecalis*, solubility, bond strength, and marginal adaptation to canal dentin between the nanofiller-incorporated BC sealer and the control sealer.

## MATERIALS AND METHODS

### Study design, consent, and inclusion criteria

The current comparative lab-based investigation followed the checklist for reporting *in-vitro* studies (CRIS) guidelines. The analyses centered on the dependent variable, with the experimental groups serving as the independent variable. The study was approved by the Ethical Committee of King Saud University under IRB # FC-5676-24. Written consent was taken from the patient whose teeth were used in the present study for experimental purposes. Forty-four human lower premolars that were extracted due to orthodontic or periodontal reasoning were utilized. To maintain consistency within the groups, digital periapical radiographs were taken from two different angles. Teeth with fully formed apex, single canal with straight roots, absence of caries no prior endodontic treatment, and no pathological modifications were included. Teeth that exhibited internal root resorptions, calcifications, fractures, or had more than one canal per root were excluded. All samples were scaled of any attached tissue and calculus using an ultrasonic scaling device (Newtron Booster, Satelec-Acteon, Merignac, France). This was followed by submerging them in 1% chloramine T solution for 24 h for disinfection (*Al-Kheraif et al., 2022*; *Alkahtany et al., 2021*).

### Nanoparticles characterization

NPs in the present study were obtained commercially (Sigma-Aldrich GmbH, Berlin, Germany) and sputter-coated using gold (Electron Microscopy Sciences, Hatfield, PA, USA) for 3 min. These NPs were then observed under a scanning electron microscope (SEM) (Mira3-XMU model–Brisbane, Queensland, Australia) at 20 KeV to identify the surface morphology. Energy-dispersive X-ray spectroscopy (EDX) (JED-2300 Analysis Station, JEOL, Japan) was used for elemental analysis (*Alfawaz et al., 2020*; *Aljamhan et al., 2021a*).

### Preparation of the NPs-modified BC sealer

The root canal sealer utilized in the existing exploration was the commercially available Total Fill BC sealer (FKG, Berlin, Germany). A pre-blended substance that was modified using different NPs. All the NPs were obtained commercially (Sigma-Aldrich GmbH, Berlin, Germany). The commercially obtained solid NPs in the concentration of 2 wt% were injected and mixed into a sealer using a precisely measured syringe and a micropipette.
## Sample size calculation, cleaning, and shaping of the canal

By using the WHO sample size calculator, the following were the calculations; confidence level = 95%, absolute precision required = 9%, population mean = 8.40, population standard deviation = 0.14, sample size = 11 cases in each group (*Al Ahdal et al., 2020*; *Habib et al., 2021*). The coronal section of the samples was sectioned using a diamond saw (Auto-Instrument Co., Shenyang, China) along with constant water irrigation till the cementoenamel junction to standardize the root length to 15 mm. Subsequently, endodontic treatment was initiated, and the final canal length was established using # 15 K file (Dentsply-Maillefer, Ballaigues, Switzerland) till anatomic foramen. The final working length (WL) was kept one mm shorter than the recorded length. For disinfection purposes, 2 ml 5.25% NaOCl solution (Cloraxid; Cerkamed, Stalowa Wola, Poland) was administered using a 30-gauge needle after each file till final cleaning and shaping. The root canals were then finally prepared utilizing a Protaper Gold rotary instrument (Dentsply Maillefer) starting with file S1 till F2 as a final finishing file. All the teeth were subsequently distributed into four groups based on the use of modified and unmodified sealers used for root canal obturation (*n* = 11) (*Al Deeb et al., 2020*; *Aljamhan et al., 2021b*).

*Group 1 (Control):* In this group, an unmodified Totalfill BC sealer was used to obturate the canal. Using paper points, the instrumented canals were dried. A single cone gutta percha size F2 along with sealer was used to fill the canal. To ensure that the obturation was adequate, radiographs of every obturated tooth were taken at two different angulations (*Alrahlah et al., 2020*). *Group 2 (AgNP-modified BC sealer):* In this group, AgNP-modified BC sealer was used to obturate the canal in the same way as Group 1. *Group 3 (CHNP-modified BC sealer):* In this group, CHNP-modified BC sealer was used to obturate the canal in the same way as Group 1. *Group 4 (HApNP modified BC sealer):* In this group, HApNP modified BC sealer was used to obturate the canal in the same way as Group 1. To ensure a complete setting, specimens were incubated (Thermo Fisher Scientific, Waltham, MA, USA) for 24 h at 37 °C and 95% humidity.

## Solubility testing of BC sealers

Solubility ($S$) was assessed following the International Standards Organization (ISO) 6876 specification. Five samples of each sealer in measurements of 5 ± 0.1 mm diameter and 2 ± 0.1 mm height were prepared using a stainless-steel ring mold and subsequently incubated at 37 °C in 100% humidity. The discs were extracted from the molds and were subjected to three weight measurements utilizing a precision balance (Ohaus Corp. Pine Brook, NJ USA). The initial mass ($I$) was documented as the average weight. The same discs were then subsequently immersed in 50 mL of distilled water set at a temperature of 37 °C and humidity of 95% for 24 h. Following that, discs were dried using absorbent paper, and positioned in a dehumidifying chamber until the mass reached stabilization and re-evaluated for their weights three additional times, and the mean weight was documented as the final mass ($F$) (*Abu Zeid & Alnoury, 2023*). The $S$ of all investigated sealers was calculated using the formula

$$S = [(I - F)/I] \times 100.$$

### Testing antibacterial property

The antimicrobial efficiency of the modified and unmodified sealers was assessed against *E. faecalis* (ATCC 29212) using an agar diffusion test (ADT). For this, 200 μL of the bacterial suspension, containing approximately $10^{\char`\^}6$ cells, was utilized. The suspension was inoculated onto agar plates containing BHI broth. Perpendicular wells with a diameter of five mm were bored and subsequently filled with each material. The plates underwent incubation at a temperature of 37 °C for 24 h. The samples were analyzed for bacterial proliferation through the assessment of the inhibition halo present on the agar medium. Repetitions were conducted to evaluate the test and the outcome of the specified material after 24 h (*Castillo-Villagomez et al., 2022*).

### Root slicing of specimens following PBS testing

The roots of 40 samples were divided into three segments, each measuring two mm in height using a diamond saw under constant water irrigation. The root slices (cervical, middle, and apical) were subsequently affixed onto an acrylic block measuring 1.5 × 1.5 mm. The PBS was assessed utilizing a universal testing machine (UTM) (Instron 3367; Instron Co., Canton, China). The cylindrical piston having a diameter of 0.7 mm and 0.4 mm and a height of four mm was used to apply continuous compressive force from the apical to the cervical direction at a speed of one mm/min until the bond between the sealer and root dentin failed. The PBS was measured in MegaPascals (MPa) by dividing the force in Newtons by the area of the bonded interface using the following formula (*Alkhudhairy et al., 2019*; *Vohra et al., 2020*)

$$\text{Bond Strength (Mpa)} = \frac{\text{Load In Newton}}{\text{Area of Bonded Surface}}.$$

### Failure analysis

The failure mode was assessed using a stereomicroscope (Leica Microsystems, Wetzlar, Berlin, Germany) at a magnification of 25×. Failures were categorized as adhesive, cohesive, and admixed.

### Marginal adaptation *via* SEM

For marginal adaptation, one sample from each group was sectioned vertically with a hard tissue microtome. Samples were fixed to an aluminum stub, underwent sputter coating with gold, and were analyzed under SEM. The maximum gap width, characterized as the largest distance between the root filling with the root dentin, was assessed visually by various examiners at the apical-3rd to mitigate any possible bias (*Enggardipta et al., 2020*; *Habib et al., 2021*).

### Statistical analysis

The normality of the data was assessed using Kolmogorov–Smirnov Test. SPSS Version 23 (IBM, Chicago, IL, USA) was used for data analysis. The results of sealer solubility, antibacterial efficacy, and PBS were assessed statistically using one-way ANOVA and *post hoc* Tukey multiple comparison tests. $p < 0.05$ for a level of significance.

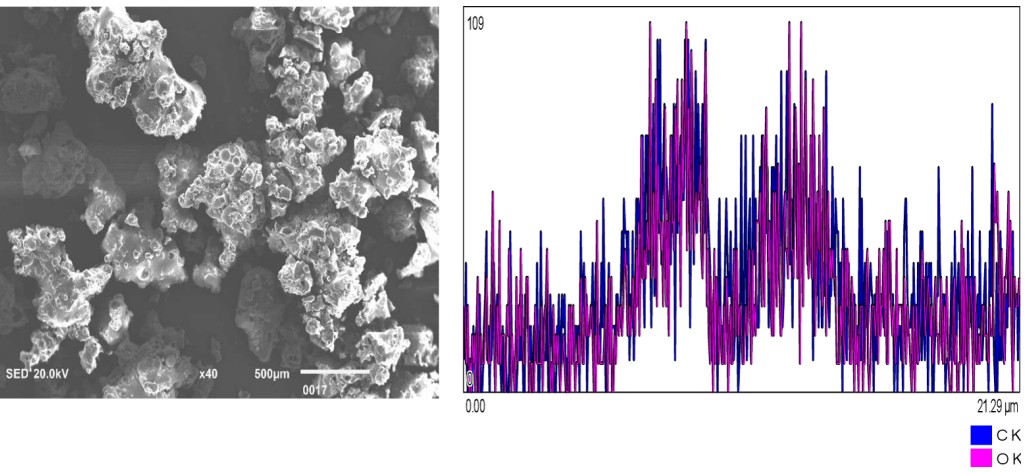

**Figure 1** **Chitosan nanoparticles of irregular shape.** EDX shows the elemental composition of carbon (C) and Oxygen ($O_2$).

## RESULTS

### Surface characterization of NPs and EDX

Figure 1: Chitosan nanoparticles exhibit an irregular morphology. Energy-dispersive X-ray (EDX) analysis reveals the elemental composition predominantly consists of carbon (C) and oxygen ($O_2$). Figure 2: Hydroxyapatite nanoparticles demonstrate agglomeration with a heterogeneous distribution of particle sizes. EDX analysis identifies calcium (Ca), oxygen ($O_2$), phosphorus (P), and carbon (C) as the major constituents. Figure 3: Silver nanoparticles display a spherical morphology and show signs of agglomeration. The major elements detected through EDX include carbon (C), oxygen ($O_2$), phosphorus (P), and silver (Ag).

### Solubility

Solubility assessment of BC root canal sealer after modifying using various nanoparticles is demonstrated in Table 1. Group 1 (Control) demonstrated the highest solubility rate. In contrast, sealers modified with NPs exhibited significantly reduced solubility in comparison to the control group yet comparable to each other ($p > 0.05$) (Fig. 4).

### Antimicrobial testing

The antimicrobial potency of different nanoparticle-modified BC sealers against *E. faecalis* is presented in Table 2. All the modified sealers exhibited the largest zone of inhibition with no significant difference *i.e.*, AgNPs modified BC sealer ($4.413 \pm 1.3$ mm), CHNPs modified BC sealer ($4.336 \pm 1.5$ mm) and HApNPs modified BC sealer ($4.321 \pm 1.0$ mm) ($p > 0.05$). Whereas the unmodified control group presented the smallest zone of bacterial inhibition ($3.343 \pm 1.2$ mm) significantly lower than the modified sealer groups ($p < 0.05$) (Fig. 5).

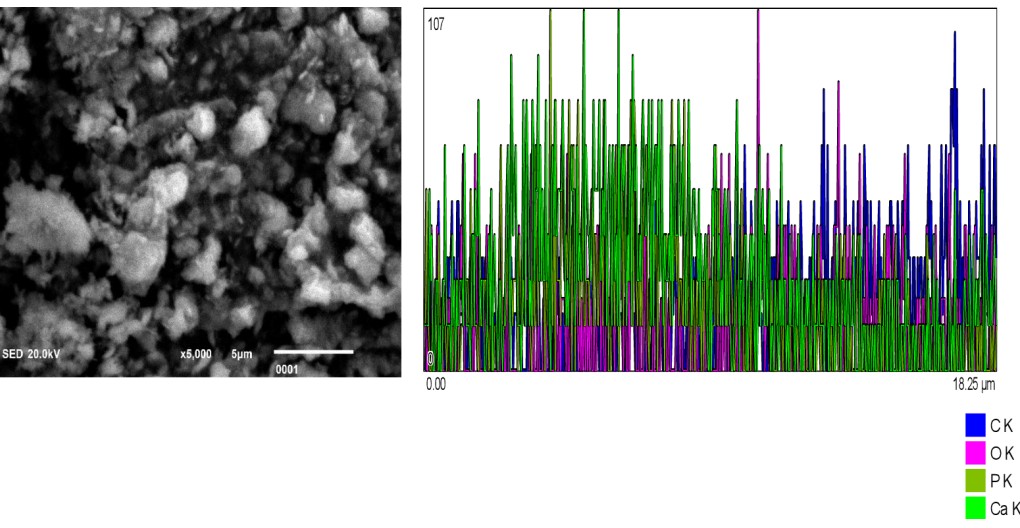

**Figure 2 Hydroxyapatite NPs showing agglomeration with heterogeneous particle size.** EDX analysis shows Calcium (Ca), oxygen (O2), Phosphorus (P), and Carbon (C) as major elements.

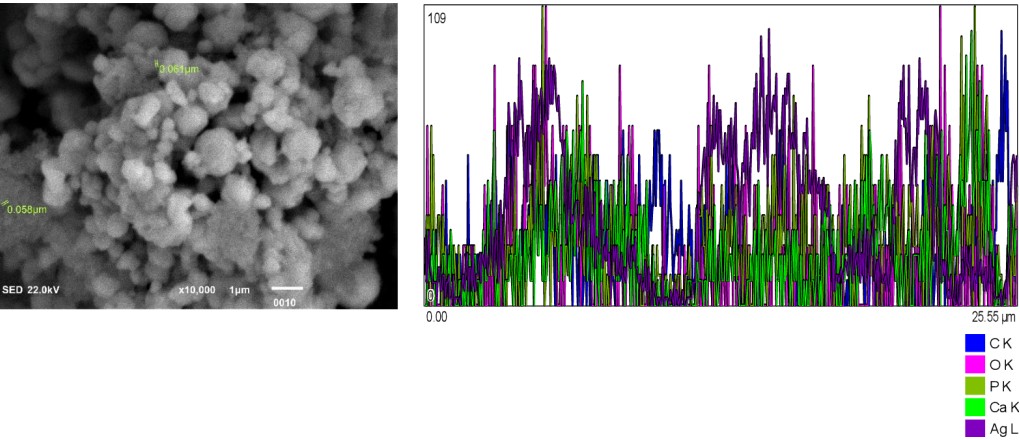

**Figure 3 Silver NPs spherical shaped and agglomerated.** Major constituents *via* EDX show Carbon (C), Oxygen (O2), Phosphorus and silver (Ag).

## PBS assessment

PBS of BC root canal sealers modified with different nanoparticles are demonstrated in Table 3. The strongest PBS was presented by a cervical third of Group 4 (HApNPs + BC sealer) specimens. However, the lowest bond scores were witnessed in the apical third of Group 1 (BC sealer) samples. Comparative analysis recognized that Group 2 (AgNPs + BC sealer), Group 3 (CHNPs + BC sealer) and Group 4 exhibited no significant disparity in their bond scores ($p < 0.05$). Nevertheless, Group 1 presented significantly lower scores than other tested groups ($p < 0.05$). Comparison within different sections of the same group discovered that the cervical and middle slices of the obturated roots established no

**Table 1  Solubility assessment of bioceramic (BC) root canal sealers modified using different nanoparticles.**

| Experimental groups | Mean ± SD (%) | *p*-value[*] |
|---|---|---|
| Group 1: BC sealer | 6.144 ± 0.39[a] | |
| Group 2: AgNPs + BC sealer | 3.993 ± 0.22[b] | *p* > 0.05 |
| Group 3: CHNPs + BC sealer | 3.988 ± 0.25[b] | |
| Group 4: HApNPs + BC sealer | 3.884 ± 0.15[b] | |

Notes.
Silver nanoparticles (AgNPs), Chitosan nanoparticles (CHNPs), Hydroxyapatite nanoparticles (HApNPs).
*ANOVA.
Different superscript lower-case alphabets denote statistically significant differences ($p < 0.05$) (*post hoc* Tukey).

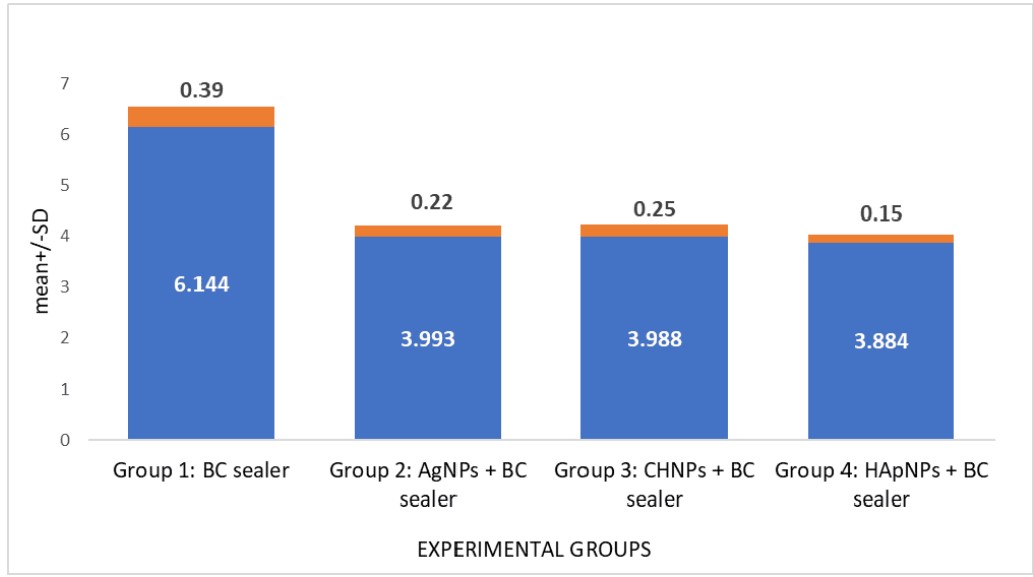

**Figure 4  Solubility evaluation of BC sealer modified with different nanoparticles.**

**Table 2  Antimicrobial assessment of BC root canal sealers modified with different nanoparticles against *E. faecalis*.**

| Experimental groups | Inhibition zone | *p*-value |
|---|---|---|
| Group 1: BC sealer | 3.343 ± 1.2[a] | 0.002 |
| Group 2: AgNPs + BC sealer | 4.413 ± 1.3[b] | 0.462 |
| Group 3: CHNPs + BC sealer | 4.336 ± 1.5[b] | 0.402 |
| Group 4: HApNPs + BC sealer | 4.321 ± 1.0[b] | 0.426 |

Notes.
Silver nanoparticles (AgNPs), Chitosan nanoparticles (CHNPs), Hydroxyapatite nanoparticles (HApNPs).
Different superscript lower-case alphabets denote statistically significant differences (*post-hoc* Tukey).

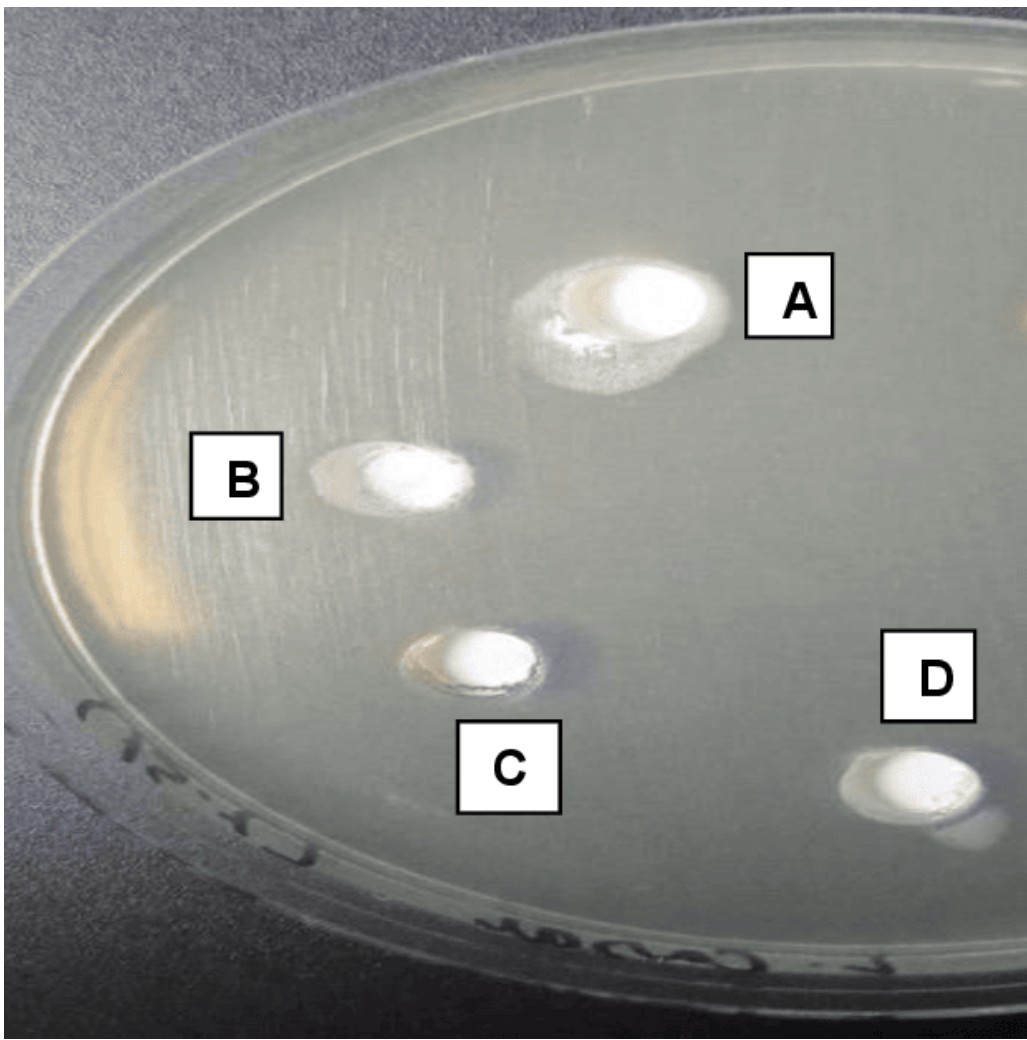

**Figure 5** **Agar diffusion test showing A, B, and D as experimentally modified sealers showing a high area of diffusion (inhibition halo).** (C) The control group with no sealer modification demonstrating the area of diffusion (Inhibition halo) less than the experimental groups.

significant difference in their PBS ($p > 0.05$). Nonetheless, apical sections showed markedly reduced bond scores ($p < 0.05$) (Fig. 6).

### Fracture pattern assessment

Results indicated that specimens from Groups 2, 3, and 4 predominantly exhibited cohesive fractures. However, adhesive fractures were most observed in Group 1 samples (Fig. 7).

### Marginal adaptation of sealer to radicular dentin at apical-third *via* SEM

Figure 8A: SEM illustrates a persistent gap between the unmodified BC sealer and canal dentin, which remains uniform across the entire interface. Additionally, instances of tearing within the sealer are also observed. Figure 8B: The SEM image of the BC sealer modified with silver reveals a pronounced gap between the canal dentin and the silver-modified

**Table 3  Push out bond strength (PBS) of BC root canal sealers to the canal dentin modified with different nanoparticles.**

| Experimental groups | Mean ± SD Coronal | Mean ± SD Middle | Mean ± SD Apical | *p*-value |
|---|---|---|---|---|
| Group 1: BC sealer | 7.28 ± 0.12*λ | 7.01 ± 0.10*λ | 6.50 ± 0.09***Λ | |
| Group 2: AgNPs + BC sealer | 9.36 ± 0.11**λ | 9.23 ± 0.13**λ | 7.14 ± 0.08***Λ | *p* > 0.05 |
| Group 3: CHNPs + BC sealer | 9.46 ± 0.07**λ | 9.39 ± 0.10**λ | 7.23 ± 0.06***Λ | |
| Group 4: HApNPs + BC sealer | 9.63 ± 0.12**λ | 9.49 ± 0.12**λ | 7.26 ± 0.07***Λ | |

**Notes.**
Silver nanoparticles (AgNPs), Chitosan nanoparticles (CHNPs), Hydroxyapatite nanoparticles (HApNPs).
*,**,*** denote statistically significant differences within the same column (*p* < 0.05).
Λ, λ denote statistically significant differences within each row (*p* < 0.05).

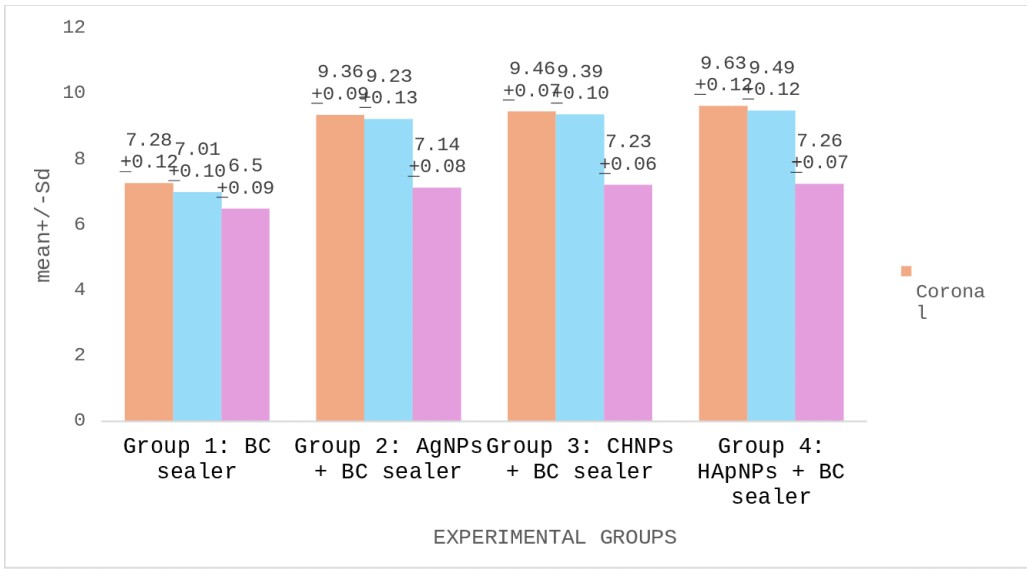

**Figure 6  Push out bond strength (PBS) of BC root canal sealers to the canal dentin modified with different nanoparticles.**

BC sealer. Nonetheless, areas of intimate contact between the sealer and dentin are also evident. Figure 8C: SEM analysis of the BC sealer modified with chitosan demonstrates a strong interface between the dentin and the chitosan-modified sealer, with no observable gap. Figure 8D: The SEM image presents the interface of the BC sealer modified with hydroxyapatite nanoparticles. The interface exhibits uneven characteristics, characterized by significant erosion and cracks in the sealer, as well as a gap between the sealer and dentin.

## DISCUSSION

The contemporary study aims to evaluate the antimicrobial and mechanical impact of AgNPs, CHNPs, and HApNPs added to a BC root canal sealer. It was assumed that there would be no significant disparity in the antimicrobial effectiveness against *E. faecalis* when nanofillers modified BC sealer was used as compared with unmodified root canal

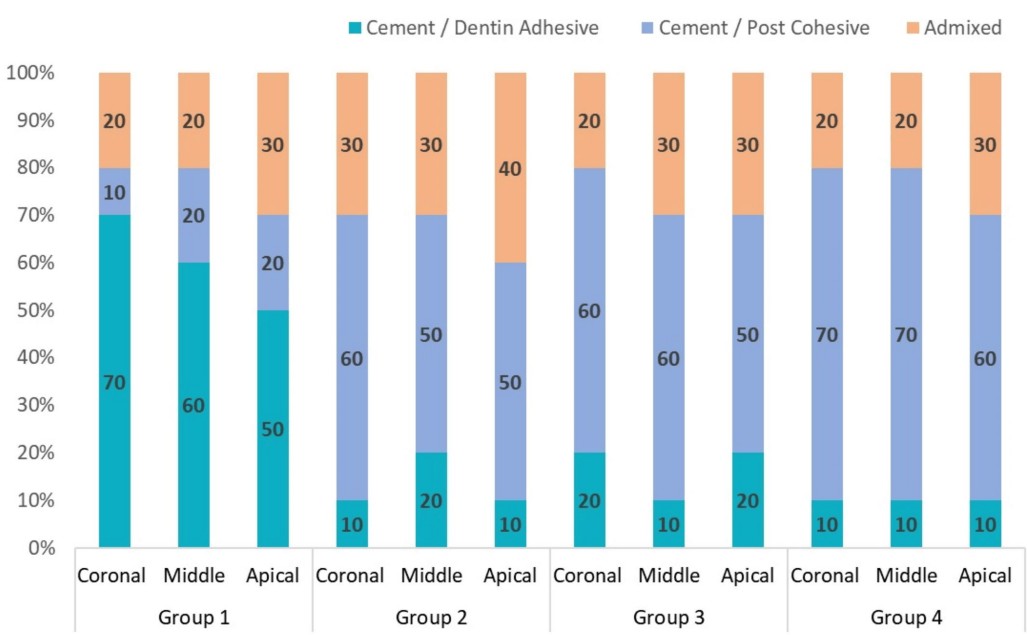

**Figure 7** Percentage of modes of failure in each group.

sealer. Additionally, it was also projected that the solubility, bond strength, and marginal adaptation at the apical-third of NPs incorporated BC sealer to the canal dentin will be comparable to the control. It was observed that both the proposed hypotheses were entirely rejected as all the modified BC sealers showed better antimicrobial potency, lower solubility, increased PBS, and better sealer adaptation at apical third compared to the control.

Endodontic treatment success requires complete bacterial elimination from the canal (*De Oliveira et al., 2017*). The results of the current investigation have shown that all NP-modified sealers exhibited significantly greater antibacterial efficacy compared to unmodified BC sealers. Specifically regarding silver nanoparticles (AgNPs), it has been noted that silver ions are released from AgNPs, which can selectively target various sites on bacterial cells (*Prabhu & Poulose, 2012*; *Wang et al., 2022*). These silver ions possess the ability to get attached to the bacterial cell membrane and affect the transport of essential ions *via* increased membrane permeability (*Wakshlak, Pedahzur & Avnir, 2015*). Moreover, silver ions interact with sulfhydryl groups in proteins and DNA, resulting in alterations in hydrogen bonding and disruptions to the respiratory chain (*Chandak et al., 2021*). However, it should be kept in consideration that AgNPs must be used with caution due to their toxicity, which is dependent on their concentration (*Kim et al., 2009*).

The antimicrobial potency of CHNPs against *E. faecalis* can be accredited to the quantum-size effect, which enhances the interaction between CHNPs and bacterial cells (*Moukarab, 2020*). Previous research has established that CHNPs with high deacetylation and low molecular weight exhibit enhanced antimicrobial activity (*Ong et al., 2017*). The increased deacetylation of chitosan facilitates a greater release of free amino groups, which interact with bacterial membranes, thereby augmenting its antibacterial

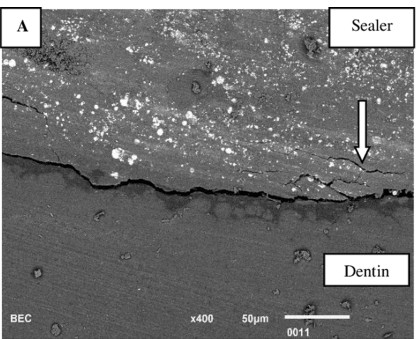

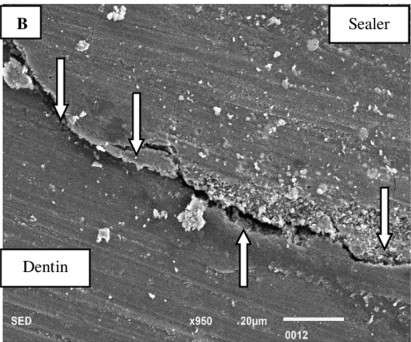

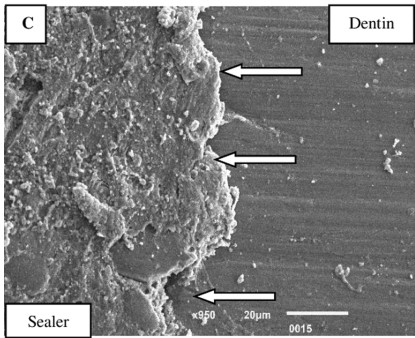

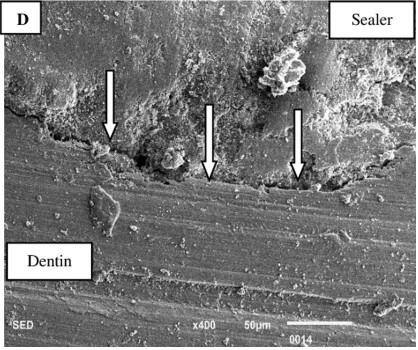

**Figure 8** **(A) A constant gap is seen between unmodified BC sealer and canal dentin *via* SEM.** The gap is constant throughout the entire interface at apical-third. Tearing of sealer is also observed arrow above. (B) Bioceramic sealer modified with silver. Arrow below indicates a gap between the canal dentin and BC sealer modified with silver. Areas of intimate contact between sealer and dentin can also be seen *via* SEM (continued on next page…)

**Figure 8 (... continued)**
at the apical-third (arrow above). (C) SEM of bioceramic sealer modified with chitosan. Arrow indicates strong interface between dentin and chitosan modified sealer. No gap is observed between the interfaces at the apical third. (D) SEM image showing interface between bio ceramic sealer modified with hydroxyapatite nanoparticles. Uneven interface with erosion and cracks in sealer. Arrows show gap between the sealer and apical-third of dentin.

effectiveness (*Rampino et al., 2013*). Moreover, HApNP-filled sealers demonstrated significant antimicrobial activity against *E. faecalis*. The antibacterial properties of HApNPs are linked to the surface charge of the microorganisms, with Gram-positive bacteria being more susceptible to hydroxyapatite. Additionally, HApNPs produce reactive oxygen species (ROS), including hydroxide ions ($OH^-$), hydrogen peroxide ($H_2O_2$), and superoxide ions ($O_2^-$), which possess bactericidal properties, thereby enhancing the elimination of bacteria (*Sobierajska et al., 2018*; *Song & Ge, 2019*).

The incorporation of nanofillers acts as stress absorbers, thereby enhancing load-bearing capacity and increasing bond strength. In terms of PBS, it was found that HApNP-filled BC sealers exhibited the highest bond integrity scores with root dentin. This finding can be supported by laboratory analyses conducted by *Witjaksono et al. (2007)* which demonstrated a correlation between finer particle sizes and lower average leakage values of root canal sealers. Furthermore, *Toledano et al. (2020)* documented the effects of using HApNPs in combination with $ZnO_2$ on microhardness, noting that HApNPs can enhance the hardness of dentin. The authors also suggested that the observed outcomes may be attributed to the remineralization capacity of the NPs (*Toledano et al., 2020*). Moreover, the integration of AgNPs and CHNPs into BC sealers also resulted in augmented PBS. The addition of these particles, characterized by an increased surface area and superior wetting properties, likely contributes to improved surface contact and enhanced bond strength of the sealer, leading to exceptional performance (*Afkhami et al., 2023*). Additionally, the distribution of NPs as reinforcing fillers can improve the mechanical properties of the sealer, augmenting its tensile and shear strength (*Viapiana et al., 2014*). A laboratory study by *Harishma et al., (2024)* assessed the PBS of epoxy resin and calcium silicate-based sealers incorporating CHNPs, reporting satisfactory results. However, there is a lack of data regarding the impact of these NPs on the bond integrity of BC sealers, indicating a need for further investigation.

Solubility refers to the loss of material when immersed in water. The current literature includes several studies on the solubility of calcium silicate-based sealers (*Marin-Bauza et al., 2010*; *Resende et al., 2009*; *Versiani et al., 2016*). One notable disadvantage of these sealers is that their solubility is higher compared to traditional epoxy-based sealers (*Gandolfi, Siboni & Prati, 2016*; *Lim et al., 2020*). For successful long-term endodontic treatment, root canal sealers must exhibit low solubility, with the mass fraction not exceeding 3%, as stipulated by ISO standards (*Baghdadi et al., 2020*). In the present investigation, TotalFill BC Sealer (unmodified) demonstrated significantly higher solubility at 6.14%. In contrast, all sealers modified with NPs exhibited lower solubility compared to the control group. This finding aligns with the results reported by *Baghdadi et al. (2020)*.

The evaluation of the marginal adaptation of BC sealer to dentin was conducted using SEM. The results of this study show that the inclusion of CHNPs in BC sealer exhibits enhanced compatibility with root dentin, surpassing the performance of BC used alone or in conjunction with AgNPs and HApNPs. It was clarified that CHNPs possess hydrophilic properties, which enhance the interaction between BC sealer and root canal dentin (*Kmiec, 2017*; *Verma et al., 2022*). However, both AgNPs and HApNPs exhibit hydrophobic properties, preventing them from making close contact with the hydrophilic dentin (*Habib et al., 2021*). The findings of the existing study indicate that the marginal adaptation of sealer does not affect the bond strength outcomes achieved.

The outcomes obtained in the existing *in vitro* analysis show potential, however, it is imperative to recognize certain limitations inherent to this research. The study utilized a lab-based model; therefore, results cannot be directly applied to clinical scenarios. The antimicrobial efficacy of nanoparticle-modified sealers in culture is influenced by various factors, such as nanoparticle size, shape, concentration, stabilization, charge, and surface functionalization properties. Furthermore, monospecies of bacteria were utilized to establish antimicrobial potency whereas endodontic infection is a multispecies infection. Additionally, EDX microanalysis presents certain limitations, including challenges in detecting light elements and an X-ray detection limit of approximately 0.1% depending on the element. Therefore, some authors propose a qualitative and quantitative overview by SEM-EDS followed by a more precise inductively coupled plasma optical emission spectroscopy (ICP-OES). Other mechanical testing of modified sealers with different NPs should also be assessed.

## CONCLUSION

The incorporation of silver, chitosan, and hydroxyapatite nanoparticles into BC root canal sealers resulted in enhanced antibacterial efficacy, improved push-out bond strength, and reduced solubility compared to unmodified BC canal sealers. Furthermore, the marginal adaptation of the sealer at the apical third is not directly correlated with bond strength.

### Funding
This work was supported by the Researchers Supporting Program (RSPD) at King Saud University through project number (RSPD2024R815), Riyadh, Saudi Arabia. The funders had no role in study design, data collection and analysis, decision to publish, or preparation of the manuscript.

### Grant Disclosures
The following grant information was disclosed by the authors:
Researchers Supporting Program (RSPD) at King Saud University: RSPD2024R815.

### Competing Interests
The authors declare there are no competing interests.

## Author Contributions

- Basil Almutairi conceived and designed the experiments, performed the experiments, analyzed the data, authored or reviewed drafts of the article, and approved the final draft.
- Fahad Alkhudhairy conceived and designed the experiments, performed the experiments, analyzed the data, prepared figures and/or tables, and approved the final draft.

## Human Ethics

The following information was supplied relating to ethical approvals (*i.e.*, approving body and any reference numbers): The Ethical Committee of King Saud University under IRB # FC-5676-24.

## Data Availability

The raw data is available in the Supplemental File.

## Supplemental Information

Supplemental information for this article can be found online at http://dx.doi.org/10.7717/peerj.18840#supplemental-information.

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
