# Peer review of "Nanoparticles modified bioceramic sealers on solubility, antimicrobial efficacy, pushout bond strength and marginal adaptation at apical-third of canal dentin"

_PeerJ, doi:10.7717/peerj.18840_

## Round 0.1 · original submission · Major Revisions

· Academic Editor

Major Revisions

Dear authors,

Thank you for submitting your manuscript, which explores a relevant and modern topic. However, the study requires significant revisions to improve clarity and scientific rigor. Details on the preparation and modification of sealers with nanoparticles, including material properties and methods, are insufficient and must be expanded. The rationale for the sample size, study design, and null hypothesis needs clarification. Statistical analyses, particularly for antimicrobial testing and PBS results, must be reviewed to resolve discrepancies in p-values and data interpretation. Additionally, tables and figures should be optimized, and the lengthy discussion condensed for better focus. Addressing these points will strengthen the manuscript and improve its suitability for publication.

Reviewer 1 ·

Basic reporting

In the abstract, some abbreviations such as “NPs” “BC sealer” were the first time appeared, the full name should be given.
In the instruction, line 45, line 48, line 70, “Root canal treatment (RCT) is the most frequently performed dental procedure” “due to their excellent biocompatibility” , should be modified, Avoidance of using “most” “excellent” as such in scientific presentation should be noted. In line 90, it seemed to be more appropriate to described as “non-inferior to that control” rather than “comparable”.
In line 224, “p” should be written in italics and before the full stop. In the result section, the paragraph of “solubility” form line 228 to 233, consider using short sentences, the current description is confusing. More illustration of the fracture pattern assessment should be included.
It may be better to turn table 2 and table 3 into figures.
The current discussion is too lengthy, a more concise one is preferred.

Experimental design

no comment

Validity of the findings

no comment

Reviewer 2 ·

Basic reporting

I believe that the topic of this article is modern and engaging, Particularly with regard to the subject of modification with NPs.
The article is well-written both linguistically and grammatically.
The references are sufficient and well-written. However, I found in introduction in the line " However, these sealers are not particularly effective against Enterococcus faecalis" this sentence needs a reference.
I think article structure, tables and Raw data shared are suitable.
This study lacks some figures of the applied push-out bond strength test , images of the materials, and visuals illustrating the process of modifying the sealer with NPs materials.
The null-hypothesis is missing.

Experimental design

I believe the issue with this study is that it examined many aspects superficially instead of focusing on a specific subject.
The section of 'Preparation of the NPs-modified BC sealer" is overly brief and requires much more details, Particularly regarding how these materials are obtained in a NPs form, as well as the nature of these materials—whether they are solid, liquid, or paste-like—and their consistency. It is necessary to provide a detailed explanation of the method used to modify the BC sealer with these materials.
One of the issues with this study is the sample size. How was the sample size calculated, and which studies were referenced to determine the sample size?
Considering that the sample size is relatively very small for all the procedures studied.

Validity of the findings

What is the current power of this study to estimate the validity of the findings?
In the section of "Antimicrobial testing" in the line 225-226 the author mentioned that " the unmodified control group presented the smallest zone of bacterial inhibition significantly different from the modified sealer groups" but in the table 1 the p value was > 0.05 in all groups which means that there was no statistically differences between groups.
It would be better to include the p-value in all the statistical tables.

Reviewer 3 ·

Basic reporting

Considerations:

Title:
It is too long. Please review it. Some words are repetitive. It is confusing the term “sealer solubility.” Is it not just “solubility” as the authors already mentioned the term “sealer”?

Abstract:
- In the item “introduction”, no background is reported, only the objective of the study. So, is it not proper to write “objective” instead of “introduction”?
- The authors should standardize the sequence of the dependent variables in both the title and objective of the study. Additionally, the same terms should be used in the whole text.
- In the “results”, the authors should report the data following the same sequence of the described dependent variables.

Material and Methods:
- The authors should state whether the extracted premolars are upper or lower.
- The item “inclusion criteria” presents information that should be described in a separate item “study design”. Additionally, provide the characteristics of the study, such as the independent and dependent variables.
- The authors did not provide enough details about how the sealers were manipulated and how the sealers were modified with each nanoparticle tested.
- What do the authors mean by “specimens sectioning”? Is it the term to refer to the preparation for the PBS test?
- Statistical analysis: Were data submitted to the normality test? For which dependent variables the one-way ANOVA was applied? And about the pos-hoc test? Please specify it.

Results:
- Report the results with the same sequence used to describe the dependent variables. Please make it sure that in the whole text, this sequence is standardized.
- If the PBS results are described in table 3, it is not necessary to repeat all the values in a descriptive text. In addition, how is it possible to have two letters (upper and lower) if the authors used one-way ANOVA? In addition, if the authors also intended to compare the different areas (coronal, middle and apical), the statistical test selected is not proper. Please review the study, design, the variables analyzed and the statistical analysis.

Experimental design

There is no description of the study design. Please see my "basic reporting".

Validity of the findings

The authors did not use the proper statistical analysis. Please see my "basic reporting".

---

## Round 0.2 · Minor Revisions

· Academic Editor

Minor Revisions

Dear authors

Your manuscript has improved; however, it may still benefit from minor revisions to the statistical analysis.

Reviewer 1 ·

Basic reporting

The current version seems acceptable.

Experimental design

The current version seems acceptable.

Validity of the findings

The current version seems acceptable.

Additional comments

The current version seems acceptable.

Reviewer 3 ·

Basic reporting

Material and Methods
- There is a misunderstanding regarding the identification of the dependent and independent variables in the study.
The dependent variable is associated with the analyses, whereas the independent variable corresponds to the experimental groups.

Statistical analysis
- Provide the result of the normality test.
- Which type of ANOVA was used? Please specify it.

Experimental design

n/a

Validity of the findings

n/a

Additional comments

n/a

---

## Round 0.3 · accepted · Accept

· Academic Editor

Accept

Dear authors,

After thorough peer review, we are pleased to accept your manuscript (Nanoparticles Modified Bioceramic Sealers on Solubility, Antimicrobial Efficacy, Pushout Bond Strength and Marginal Adaptation at apical-third of Canal Dentin) for publication. The reviewers commend the clarity of the writing, the relevance of the findings, and the potential to improve clinical outcomes.

Congratulations on this achievement!

Reviewer 3 ·

Basic reporting

After the review, the manuscript was improved, and the corrections were performed.

Experimental design

no comment

Validity of the findings

no comment

Additional comments

no comment